# A power approximation for the Kenward and Roger Wald test in the linear mixed model

**Sarah M. Kreidler**[1☯], **Brandy M. Ringham**[2☯]\*, **Keith E. Muller**[3☯], **Deborah H. Glueck**[4☯]

**1** Department of Biostatistics and Informatics, University of Colorado Denver, Aurora, CO, United States of America, **2** LEAD Center, University of Colorado Denver, Aurora, CO, United States of America, **3** Department of Health Outcomes and Biomedical Informatics, University of Florida, Gainesville, FL, United States of America, **4** Department of Pediatrics, University of Colorado Denver, Aurora, CO, United States of America

☯ These authors contributed equally to this work.
\* brandy.ringham@cuanschutz.edu

## Abstract

We derive a noncentral $\mathcal{F}$ power approximation for the Kenward and Roger test. We use a method of moments approach to form an approximate distribution for the Kenward and Roger scaled Wald statistic, under the alternative. The result depends on the approximate moments of the unscaled Wald statistic. Via Monte Carlo simulation, we demonstrate that the new power approximation is accurate for cluster randomized trials and longitudinal study designs. The method retains accuracy for small sample sizes, even in the presence of missing data. We illustrate the method with a power calculation for an unbalanced group-randomized trial in oral cancer prevention.

**Data Availability Statement:** Source code, data and instructions for reproducing the manuscript results are available at http://github.com/SampleSizeShop/mixedPower.

## 1 Introduction

### 1.1 Motivation

Linear mixed models are widely used in biomedical research for inference in analyses with missing data. Kenward and Roger [1] described a scaled Wald statistic and null case reference distribution for tests of fixed effects in the linear mixed model. Despite the widespread use of the Kenward and Roger [1] method for data analysis, no general methods are available to calculate power for the Kenward and Roger [1] test.

Several authors have described power approximations for related tests and models. Helms [2] described a noncentral $\mathcal{F}$ power approximation for a Wald test. Helms used a different null case reference distribution than the one derived by Kenward and Roger. Stroup [3] suggested an "exemplary data" approach for calculating power for mixed models with missing data. Tu *et al.* [4, 5] developed an asymptotic power approximation based on generalized estimating equations. Shieh [6] provided non-central $\mathcal{F}$ power approximations for multivariate models with random covariates and no missing data. Chi, Glueck, and Muller [7] demonstrated that power methods for the general linear multivariate model may be used in complete, balanced, homoscedastic mixed models.

**Funding:** This study was supported by The National Institute of Dental and Craniofacial Research (www.nih.gov) in the form of a grant awarded to KEM and DHG (NIDCR 1 R01 DE020832-01A1), The National Institute of General Medical Sciences (www.nih.gov) in the form of a grant awarded to KEM and DHG (NIGMS 9R01GM121081-05), and the Office of the Director (www.nih.gov) in the form of a grant awarded to Dana Dabelea, PI (OD 5UG3OD023248-02). The funders had no role in study design, data collection and analysis, decision to publish, or preparation of the manuscript. The funders had no role in study design, data collection and analysis, decision to publish, or preparation of the manuscript.

We derive a noncentral $\mathcal{F}$ power approximation for the Kenward and Roger [1] test for a broad range of models. We use a method of moments approach [8] to form an approximate distribution of the Kenward and Roger [1] scaled Wald statistic, $F_R$, under the alternative. The reference distribution of $F_R$ under the alternative depends on the approximate moments of the unscaled Wald statistic.

The remainder of the manuscript is organized as follows. In Section 2, we introduce notation for the general linear mixed model and briefly review the methods of Kenward and Roger [1]. In Section 3, we describe a noncentral $\mathcal{F}$ power approximation for the Kenward and Roger [1] test. In Section 4, we summarize the Monte Carlo simulation study used to evaluate the power approximation. In Section 5, we demonstrate a power calculation for a longitudinal trial in oral cancer prevention. In Section 6, we provide concluding remarks.

## 2 Notation, models, and hypothesis testing

### 2.1 Notation

For $i \in \{1, \ldots, n\}$, let $\boldsymbol{a} = \{a_i\}$ denote an $n \times 1$ column vector. Furthermore, for $i \in \{1, \ldots, n\}$ and $j \in \{1, \ldots, m\}$, let $\boldsymbol{A} = \{a_{ij}\}$ indicate an $n \times m$ matrix with transpose $\boldsymbol{A}' = \{a_{ji}\}$. Let $\boldsymbol{I}_d$ be a $(d \times d)$ identity matrix. For a matrix $\boldsymbol{A} = [\boldsymbol{a}_1 \, \boldsymbol{a}_2 \ldots \boldsymbol{a}_n]$, let $\text{vec}(\boldsymbol{A}) = \begin{bmatrix} \boldsymbol{a}_1' & \boldsymbol{a}_2' & \ldots & \boldsymbol{a}_n' \end{bmatrix}'$. Define the Kronecker product of two matrices $\boldsymbol{A}$ and $\boldsymbol{B}$ as $\boldsymbol{A} \otimes \boldsymbol{B} = \{a_{ij} \boldsymbol{B}\}$ [9, Section 1.3].

Extend the direct sum operator [9, Section 1.3] to sets of arbitrarily sized matrices as follows. Let $\{\boldsymbol{A}_1, \ldots, \boldsymbol{A}_J\}$ be a set of matrices such that $\boldsymbol{A}_j$ has dimension $(r_j \times c_j)$. Let $\boldsymbol{0}_{r_i,c_j}$ be an $(r_i \times c_j)$ matrix of zeros. Define the direct sum of $\{\boldsymbol{A}_1, \ldots, \boldsymbol{A}_J\}$ as

$$\bigoplus_{j=1}^{J} \boldsymbol{A}_j = \begin{bmatrix} \boldsymbol{A}_1 & \boldsymbol{0}_{r_1,c_2} & \cdots & \boldsymbol{0}_{r_1,c_J} \\ \boldsymbol{0}_{r_2,c_1} & \boldsymbol{A}_2 & & \vdots \\ \vdots & & \ddots & \boldsymbol{0}_{r_{J-1},c_J} \\ \boldsymbol{0}_{r_J,c_1} & \cdots & \boldsymbol{0}_{r_J,c_{J-1}} & \boldsymbol{A}_J \end{bmatrix}. \tag{1}$$

For $\delta \in \{1, \ldots, (2^p - 1)\}$ and $d \in \{1, \ldots, \delta\}$, define the set $R_d$ where $R_d \subseteq \{1, \ldots, p\}$ of cardinality $1 \leq p_d \leq p$. For every $R_d$, let $\boldsymbol{D}_{p,d}$, a *deletion matrix*, be the $(p_d \times p)$ submatrix of $\boldsymbol{I}_p$ formed by keeping each row $i$ of $\boldsymbol{I}_p$ such that $i \in R_d$. For example, given a $(p \times p)$ matrix $\boldsymbol{A}$ and $R_d = \{1, 3\}$,

$$\boldsymbol{D}_{3,d} = \begin{bmatrix} 1 & 0 & 0 \\ 0 & 0 & 1 \end{bmatrix} \tag{2}$$

and

$$\boldsymbol{D}_{3,d} \boldsymbol{A} \boldsymbol{D}_{3,d}' = \begin{bmatrix} a_{11} & a_{13} \\ a_{31} & a_{33} \end{bmatrix}. \tag{3}$$

Let $E_0(u)$ and $E_A(u)$ be the expectations of the random variable $u$ under the null and alternative hypotheses, respectively. Similarly, let $\mathcal{V}_0(u)$ and $\mathcal{V}_A(u)$ indicate the variance under the null and alternative hypotheses. For random matrix variates, denote the covariance under the null and alternative hypotheses as $\mathcal{V}_0(\boldsymbol{A})$ and $\mathcal{V}_A(\boldsymbol{A})$, respectively.

Let $X \sim \mathcal{D}$ indicate that random variable $X$ follows a distribution $D$ exactly, while $X \overset{.}{\sim} \mathcal{D}$ indicates that distribution is followed approximately. Let $F \sim \mathcal{F}(v_n, v_d, \gamma)$ indicate that the random variable $F$ follows a noncentral $\mathcal{F}$ distribution [10] with numerator degrees of freedom $v_n$, denominator degrees of freedom $v_d$, and noncentrality parameter $\gamma$. For $\gamma = 0$, $F$ is said to follow a central $\mathcal{F}$ distribution, written $F \sim \mathcal{F}(v_n, v_d)$. Define $\mathcal{F}^{-1}(b; v_n, v_d, \gamma)$ such that for $0 \le b \le 1$

$$\mathcal{F}(f; v_n, v_d, \gamma) = b \Leftrightarrow \mathcal{F}^{-1}(b; v_n, v_d, \gamma) = f. \tag{4}$$

Use $Y \sim \mathcal{N}_{N,p}(M, \Xi, \Sigma)$ to indicate that the $(N \times p)$ matrix $Y$ follows a matrix Gaussian distribution, with $M$ an $(N \times p)$ matrix of means, $\Xi$ an $(N \times N)$ symmetric, positive definite column covariance matrix, and $\Sigma$ a $(p \times p)$ symmetric, positive definite row covariance matrix [9, Chapter 8]. Write $W \sim \mathcal{W}_p(N, \Sigma)$ to indicate that the $(p \times p)$ matrix $W$ follows a central Wishart distribution of dimension $p$, degrees of freedom $N$, on covariance $\Sigma$. For $\Psi = \Sigma^{-1}$, write $W^{-1} \sim \mathcal{IW}_p\{(N + p + 1), \Psi\}$ to indicate that $W^{-1}$ follows a central inverse Wishart distribution of dimension $p$, degrees of freedom $N + p + 1$, and precision matrix $\Psi$ [11, p. 111, Theorem 3.4.1].

## 2.2 The general linear mixed model

We describe the general linear mixed model for Gaussian outcomes using the notation of Muller and Stewart [9, Chapter 5]. Let $i \in \{1, \ldots, N\}$ indicate the $i$th *independent sampling unit* [9, Chapter 5]. An independent sampling unit may be a single participant, as in a clinical trial, or a group of participants, as in a cluster-randomized study. Observations from two different independent sampling units are statistically independent. Observations within an independent sampling unit may be correlated. For example, for a particpant in a longitudinal trial, repeated measurements over time will be correlated.

Let $p_i$ be the number of observations for the $i$th independent sampling unit, with $p = \max_i(p_i)$. For the $i$th independent sampling unit, let $y_i$ be the $(p_i \times 1)$ vector of observed outcomes, $X_i$ be the $(p_i \times r)$ fixed effects design matrix of rank $r$, and $e_i$ be the $(p_i \times 1)$ vector of random errors. Assume that for $i \ne j$, $e_i \perp e_j$ and $y_i \perp y_j$. Let $\Sigma_i$ be a $(p_i \times p_i)$ symmetric, positive definite matrix, with

$$e_i \sim \mathcal{N}_{p_i}(0, \Sigma_i). \tag{5}$$

Let $\beta$ be the $(r \times 1)$ vector of regression parameters. The linear mixed model for the $i$th independent sampling unit is

$$y_i = X_i \beta + e_i. \tag{6}$$

Let $n = \sum_{i=1}^{N} p_i$. Define the $(n \times 1)$ vectors $y_s = \begin{bmatrix} y_1' & y_2' & \ldots & y_N' \end{bmatrix}'$ and $e_s = \begin{bmatrix} e_1' & e_2' & \ldots & e_N' \end{bmatrix}'$. Stack the fixed effect design matrices into the $(n \times r)$ matrix

$$X_s = \begin{bmatrix} X_1' & X_2' & \ldots & X_N' \end{bmatrix}'. \tag{7}$$

Throughout, we assume that predictor values are not allowed to change within an independent sampling unit, *i.e.*, that there are no repeated covariates. In addition, we assume that all predictor values are fixed as part of the study design. The population-averaged form of the linear mixed model is

$$y_s = X_s \beta + e_s. \tag{8}$$

Define

$$\boldsymbol{\Sigma}_s = \overset{N}{\underset{i=1}{\oplus}} \boldsymbol{\Sigma}_i. \tag{9}$$

The distribution of $\boldsymbol{y}_s$ is

$$\boldsymbol{y}_s \sim \mathcal{N}_n(\boldsymbol{X}_s\boldsymbol{\beta}, \boldsymbol{\Sigma}_s). \tag{10}$$

## 2.3 Tests for fixed effects in mixed models

Let $\alpha$ be the Type I error rate. Let $\boldsymbol{C}$ be the $(a \times r)$ matrix of fixed effects contrasts. Define the $(a \times 1)$ matrix $\boldsymbol{\theta} = \boldsymbol{C}\boldsymbol{\beta}$, and let $\boldsymbol{\theta}_0$ be the $(a \times 1)$ matrix of null values. The general linear hypothesis may be stated as

$$H_0 : \boldsymbol{\theta} = \boldsymbol{\theta}_0. \tag{11}$$

In order to conduct power analysis for the general linear hypothesis in the mixed model, we must consider the target estimation method. Several estimation methods have been described for mixed models [12, Chapter 5]. Common estimation methods include restricted maximum likelihood and maximum likelihood.

Let $m$ indicate the estimation method. Let $\hat{\boldsymbol{\Sigma}}_{s,m}$ and $\hat{\boldsymbol{\beta}}_m$ be the estimates of $\Sigma_s$ and $\boldsymbol{\beta}$ obtained from method $m$. Define $\hat{\boldsymbol{\theta}}_m = \boldsymbol{C}\hat{\boldsymbol{\beta}}_m$. The Wald statistic for the linear mixed model is

$$w_m = (\hat{\boldsymbol{\theta}}_m - \boldsymbol{\theta}_0)'[\boldsymbol{C}(\boldsymbol{X}_s'\hat{\boldsymbol{\Sigma}}_{s,m}^{-1}\boldsymbol{X}_s)^{-1}\boldsymbol{C}']^{-1}(\hat{\boldsymbol{\theta}}_m - \boldsymbol{\theta}_0)/a. \tag{12}$$

The distribution of the Wald statistic is not known exactly for any $m$. Various reference distributions have been suggested for each estimation method $m$. In general, the distributions share a common form, with

$$w_m \overset{\cdot}{\sim} \mathcal{F}(v_{n,m}, v_{d,m}, \gamma_m). \tag{13}$$

Under the null hypothesis, $\gamma_m = 0$ and $w_m \overset{\cdot}{\sim} \mathcal{F}(v_{n,m}, v_{d,m})$.

## 2.4 The Kenward-Roger test for fixed effects

Kenward and Roger [1] suggested using restricted maximum likelihood estimation ($m = R$) and a scaled Wald statistic.

$$\begin{aligned} F_R &= \lambda(\hat{\boldsymbol{\theta}}_R - \boldsymbol{\theta}_0)'[\boldsymbol{C}(\boldsymbol{X}_s'\hat{\boldsymbol{\Sigma}}_{s,R}^{-1}\boldsymbol{X}_s)^{-1}\boldsymbol{C}']^{-1}(\hat{\boldsymbol{\theta}}_R - \boldsymbol{\theta}_0)/a. \\ &= \lambda w_R \end{aligned} \tag{14}$$

Kenward and Roger [1] used Taylor expansion to estimate $E_0(w_R)$ and $\mathcal{V}_0(w_R)$ from observed data. Kenward and Roger [1] substituted $E_0(w_R)$ and $\mathcal{V}_0(w_R)$ into method of moments approximations for $\lambda$ and the reference distribution of $F_R$ under the null. With $F_R \overset{\cdot}{\sim} \mathcal{F}(a, v)$,

$$\rho = \frac{\mathcal{V}_0(w_R)}{2E_0(w_R)^2}, \tag{15}$$

$$v = 4 + \frac{a+2}{a\rho - 1}, \tag{16}$$

and

$$\lambda = \frac{v}{(v-2)E_0(w_R)}.$$ (17)

## 3 Power approximation for the Kenward-Roger test in the linear mixed model

### 3.1 The approximate moments of the Wald statistic

We derive a noncentral $\mathcal{F}$ power approximation for the Kenward and Roger [1] test. The method of moments approach [8] is used to form an approximate distribution of the Kenward and Roger [1] scaled Wald statistic, $F_R$, under the alternative. The reference distribution of $F_R$ under the alternative depends on the approximate moments of the unscaled Wald statistic.

We demonstrate that the Wald statistic has an approximately noncentral $\mathcal{F}$ reference distribution under the alternative and a central $\mathcal{F}$ reference distribution under the null. The result depends on approximate distributional results for both $(\hat{\boldsymbol{\theta}}_R - \boldsymbol{\theta}_0)$ and $\boldsymbol{C}(\boldsymbol{X}_s'\hat{\boldsymbol{\Sigma}}_{s,R}^{-1}\boldsymbol{X}_s)^{-1}\boldsymbol{C}'$. Because distributional results are, in general, not available for restricted maximum likelihood estimation, we instead use distributional results based on other techniques.

Let $m = W$ indicate weighted least squares, and $m = M$ denote multivariate methods. Approximate $(\hat{\boldsymbol{\theta}}_R - \boldsymbol{\theta}_0)$ by $(\hat{\boldsymbol{\theta}}_W - \boldsymbol{\theta}_0)$, which is Gaussian, conditional on $\Sigma_s$. The term $\boldsymbol{C}(\boldsymbol{X}_s'\hat{\boldsymbol{\Sigma}}_{s,R}^{-1}\boldsymbol{X}_s)^{-1}\boldsymbol{C}'$ can be approximated by $\boldsymbol{C}(\boldsymbol{X}_s'\hat{\boldsymbol{\Sigma}}_{s,M}^{-1}\boldsymbol{X}_s)^{-1}\boldsymbol{C}'$. We show that $\boldsymbol{C}(\boldsymbol{X}_s'\hat{\boldsymbol{\Sigma}}_{s,M}^{-1}\boldsymbol{X}_s)^{-1}\boldsymbol{C}'$ is approximately Wishart. Finally, under the assumption of independence, we combine the terms to obtain an approximate $\mathcal{F}$ distribution.

**3.1.1 The conditional distribution of $(\hat{\theta}_W - \theta_0)$.** The weighted least squares estimate [12] of $\boldsymbol{\beta}$ is

$$\hat{\boldsymbol{\beta}}_W = (\boldsymbol{X}_s'\boldsymbol{\Sigma}_s^{-1}\boldsymbol{X}_s)^{-1}(\boldsymbol{X}_s'\boldsymbol{\Sigma}_s^{-1}\boldsymbol{y}_s).$$ (18)

With $\hat{\boldsymbol{\theta}}_W = \boldsymbol{C}\hat{\boldsymbol{\beta}}_W$,

$$(\hat{\boldsymbol{\theta}}_W - \boldsymbol{\theta}_0)|\boldsymbol{\Sigma}_s \sim \mathcal{N}_a\{(\boldsymbol{\theta} - \boldsymbol{\theta}_0), \boldsymbol{C}(\boldsymbol{X}_s'\boldsymbol{\Sigma}_s^{-1}\boldsymbol{X}_s)^{-1}\boldsymbol{C}'\}.$$ (19)

**3.1.2 The approximate distribution of $\boldsymbol{C}(\boldsymbol{X}_s'\hat{\boldsymbol{\Sigma}}_{s,M}^{-1}\boldsymbol{X}_s)^{-1}\boldsymbol{C}'$.** We approximate the distribution of

$$\boldsymbol{C}(\boldsymbol{X}_s'\hat{\boldsymbol{\Sigma}}_{s,M}^{-1}\boldsymbol{X}_s)^{-1}\boldsymbol{C}'$$ (20)

with a single central Wishart. The result follows from Theorems 1, 2 and 3 in A. The theorems provide an approximate distribution for a positive definite sum of potentially singular quadratic forms in independent inverse central Wishart matrices.

The accuracy of the approximation depends on the degrees of freedom of the component quadratic forms. To ensure sufficient degrees of freedom, we make the following homoscedasticity assumptions. Recall $p = \max_i(p_i)$. With $\Sigma_{max}$ a symmetric, positive definite matrix, assume $\Sigma_i \equiv \Sigma_{max}$ for all $i \in \{1, \ldots, N\}$ such that $p_i = p$. Let $N_d$ indicate the number of independent sampling units with observation pattern $R_d$. Note $N = \sum_{d=1}^{\delta} N_d$. For independent sampling units with observation pattern $R_d$, assume

$$\boldsymbol{\Sigma}_d = \boldsymbol{D}_{p,d}\boldsymbol{\Sigma}_{max}\boldsymbol{D}_{p,d}'.$$ (21)

Without loss of generality, permute the independent sampling units in Eq 8 so that

$$\mathbf{\Sigma}_s = \overset{\delta}{\underset{d=1}{\oplus}} \overset{N_d}{\underset{i=1}{\oplus}} \mathbf{\Sigma}_d. \tag{22}$$

Estimate $\mathbf{\Sigma}_s$ with

$$\hat{\mathbf{\Sigma}}_s = \overset{\delta}{\underset{d=1}{\oplus}} \overset{N_d}{\underset{i=1}{\oplus}} \hat{\mathbf{\Sigma}}_d \tag{23}$$

The following thought experiment gives reasonable approximations for the distribution of each $\hat{\mathbf{\Sigma}}_d$. All independent sampling units with observed data pattern $R_d$ have $p_d$ observations. For each $R_d$, suppose we form a complete, balanced mixed model containing only the independent sampling units with observed data pattern $R_d$. For each balanced mixed model, assume that $X_s$ includes the full time by treatment interaction. This permits recasting each balanced mixed model as an equivalent general linear multivariate model [9, Chapter 14]. For cluster randomized designs, we assume that the mixed model is recast as a two-stage model of cluster means [13, Chapter 4], a special case of the multivariate model.

For the $d$th multivariate model, let $q$ be the rank of the multivariate design matrix and $\hat{E}_d$ be the $(N_d \times p_d)$ matrix of residuals. Assume $N_d > (q + p_d + 1)$. Then an unbiased, consistent estimate of $\Sigma_d, \hat{\mathbf{\Sigma}}_{d,M}$, can be formed using known results for the multivariate model. Thus,

$$\hat{\mathbf{\Sigma}}_{d,M} = \hat{E}'_d \hat{E}_d / (N_d - q), \tag{24}$$

with distribution

$$\hat{\mathbf{\Sigma}}_{d,M} \sim \mathcal{W}_{p_d}\{N_d - q, \mathbf{\Sigma}_d / (N_d - q)\}. \tag{25}$$

Recall that in the Wald statistic (Eq 12),

$$X'_s \hat{\mathbf{\Sigma}}^{-1}_{s,M} X_s = \sum_{d=1}^{\delta} \sum_{i=1}^{N_d} X'_i \hat{\mathbf{\Sigma}}^{-1}_{d,M} X_i. \tag{26}$$

Using Eq 25 and Theorem 3 in Appendix, approximate the distribution of $X'_s \hat{\mathbf{\Sigma}}^{-1}_{s,M} X_s$ with a single inverse central Wishart,

$$X'_s \hat{\mathbf{\Sigma}}^{-1}_{s,M} X_s \overset{.}{\sim} \mathcal{IW}_r(N_*, \mathbf{\Sigma}^{-1}_*). \tag{27}$$

From the linear properties of Wishart matrices [11, p. 111, Theorem 3.4.1],

$$C(X'_s \hat{\mathbf{\Sigma}}^{-1}_{s,M} X_s)^{-1} C' \overset{.}{\sim} \mathcal{W}_a\{(N_* - r - 1), C\mathbf{\Sigma}_* C'\}. \tag{28}$$

**3.1.3 Combining $(\hat{\theta}_W - \theta_0)$ and $C(X'_s \hat{\mathbf{\Sigma}}^{-1}_{s,M} X_s)^{-1} C'$ to form an approximate $\mathcal{F}$.** We now combine $(\hat{\boldsymbol{\theta}}_W - \boldsymbol{\theta}_0)$ and $C(X'_s \hat{\mathbf{\Sigma}}^{-1}_{s,M} X_s)^{-1} C'$ as described in Sections 3.1.1 and 3.1.2 to form a Wald statistic,

$$w = (\hat{\boldsymbol{\theta}}_W - \boldsymbol{\theta}_0)' [C(X'_s \hat{\mathbf{\Sigma}}^{-1}_{s,M} X_s)^{-1} C']^{-1} (\hat{\boldsymbol{\theta}}_W - \boldsymbol{\theta}_0)/a. \tag{29}$$

We assume that $w \approx w_R$. From Eq 19, $(\hat{\boldsymbol{\theta}}_W - \boldsymbol{\theta}_0)$ is approximately Gaussian. From Eq 28, $C(X'_s \hat{\mathbf{\Sigma}}^{-1}_{s,M} X_s)^{-1} C'$ is approximately Wishart.

For conciseness of notation, write $\boldsymbol{\mu} = (\boldsymbol{\theta} - \boldsymbol{\theta}_0)$, with estimate $\hat{\boldsymbol{\mu}} = (\hat{\boldsymbol{\theta}}_W - \boldsymbol{\theta}_0)$, $\boldsymbol{W} = \boldsymbol{C}(\boldsymbol{X}_s'\boldsymbol{\Sigma}_s^{-1}\boldsymbol{X}_s)^{-1}\boldsymbol{C}'$ and $\hat{\boldsymbol{W}} = \boldsymbol{C}(\boldsymbol{X}_s'\hat{\boldsymbol{\Sigma}}_{s,M}^{-1}\boldsymbol{X}_s)^{-1}\boldsymbol{C}'$. Define $\boldsymbol{Q} = [\mathcal{V}(\hat{\boldsymbol{W}})]^{-1}\mathcal{V}(\hat{\boldsymbol{\mu}})$ and $h = \boldsymbol{\mu}'[\mathcal{V}(\hat{\boldsymbol{W}})]^{-1}\boldsymbol{\mu}$. Assume that $\hat{\boldsymbol{\theta}}_W \perp \hat{\boldsymbol{\Sigma}}_{s,M}$. The assumption rests on the following logic. If we had estimated both $\Sigma_s$ and $\boldsymbol{\beta}$ using multivariate techniques, independence would follow [14, p. 291, Theorem 8.2.2]. Applying Theorem 4 in Appendix,

$$w \stackrel{\cdot}{\sim} \{a(N_* - r + a - 2)\}^{-1}\mathrm{tr}(\boldsymbol{Q})\mathcal{F}\{n_u, (N_* - r + a - 2), \delta_u\}, \tag{30}$$

where

$$\delta_u = \frac{h\,\mathrm{tr}(\boldsymbol{Q}) + 2h^2}{\mathrm{tr}(\boldsymbol{QQ}') + 2\boldsymbol{\mu}'\{\boldsymbol{Q}[\mathcal{V}(\hat{\boldsymbol{W}})]^{-1}\}\boldsymbol{\mu}} \tag{31}$$

and

$$n_u = \delta_u h^{-1}\mathrm{tr}(\boldsymbol{Q}). \tag{32}$$

From Eq 30, we calculate $E_0(w)$, $E_A(w)$, and $\mathcal{V}_A(w)$, using standard results for central and noncentral $\mathcal{F}$ distributions [10].

## 3.2 A three-moment approximation for the distribution of the Kenward and Roger scaled Wald statistic under the alternative hypothesis

We use a method of moments approach [8] to form the approximate distribution of Kenward and Roger [1] scaled Wald statistic, $F_R$, under the alternative. The parameters of the distribution depend on the approximate Wald moments derived in Section 3.1. We approximate the distribution of the Kenward and Roger [1] statistic, $F_R = \lambda w_R$, by the distribution of $F = \lambda w$, where $F \stackrel{\cdot}{\sim} \mathcal{F}(a, v, \gamma)$. Thus

$$F_R \stackrel{\cdot}{\sim} \mathcal{F}(a, v, \gamma). \tag{33}$$

To obtain values for $\lambda$, $v$, and $\gamma$ under the alternative, we match three moments, setting

$$E_A(F) = E_A(\lambda w), \tag{34}$$

$$\mathcal{V}_A(F) = \mathcal{V}_A(\lambda w), \tag{35}$$

and

$$E_0(F) = E_0(\lambda w). \tag{36}$$

With

$$\rho = \frac{\mathcal{V}_A(w)}{2\{E_0(w)\}^2}, \tag{37}$$

we obtain

$$\lambda = \frac{v}{(v-2)E_0(w)}, \tag{38}$$

$$v = 4 + \frac{2(a + 2\gamma) + (a + \gamma)^2}{\rho a^2 - a - 2\gamma}, \tag{39}$$

and

$$\gamma = a \left\{ \frac{E_A(w)}{E_0(w)} - 1 \right\}. \tag{40}$$

When $\gamma = 0$, Eq 39 reduces to

$$v = 4 + \frac{a+2}{a\rho - 1}, \tag{41}$$

which shares the same form as the result obtained by Kenward and Roger (Eq 16). The exact values of $\rho$, and hence $v$, will differ due to the disparate techniques used to obtain moments for the Wald statistics, $w$ and $w_R$.

## 3.3 Power calculation for the Kenward and Roger test

We calculate power for the Kenward and Roger test as follows. Define $\alpha$, $\Sigma_{max}$, $\boldsymbol{\beta}$, $\boldsymbol{C}$ and $\boldsymbol{\theta}_0$. For $i \in \{1, \ldots, N\}$, specify $\boldsymbol{X}_i$ and $R_d$. Calculate $a$, $v$, and $\gamma$ as described in Section 3.2. Form the reference distribution of $F_R \overset{.}{\sim} \mathcal{F}(a, v, \gamma)$. Using the approximate reference distribution of $F_R$ under the null, $F_R \overset{.}{\sim} \mathcal{F}(a, v, 0)$, find the critical value

$$f_{crit} \approx \mathcal{F}^{-1}(1 - \alpha; a, v, 0). \tag{42}$$

Finally, using the approximate reference distribution of $F_R$ under the alternative, $F_R \overset{.}{\sim} \mathcal{F}(a, v, \gamma)$, calculate power as

$$\text{Power} \approx 1 - \mathcal{F}(f_{crit}; a, v, \gamma). \tag{43}$$

# 4 Simulation study

## 4.1 Methods

We compared approximate power values, calculated as in Section 3.3, with empirical power for two types of study designs: unbalanced, cluster randomized trials and longitudinal studies with known dropout patterns. Approximate power was calculated using our *mixedPower* package for R version 4.0.2 [15].

Empirical power was calculated by Monte Carlo simulation in SAS [16, version 9.4]. We defined $\alpha$, $\Sigma_{max}$, $\boldsymbol{\beta}$, $\boldsymbol{C}$ and $\boldsymbol{\theta}_0$. For $i \in \{1, \ldots, N\}$, we specified $\boldsymbol{X}_i$ and $R_d$. We generated 10, 000 replicates of $\boldsymbol{e}_s$ and computed $\boldsymbol{y}_s$ as in Eq 8. For each replicate, we tested the linear contrast $\boldsymbol{C}$ using SAS PROC MIXED with the DDFM = KenwardRoger flag to request Kenward and Roger [1] denominator degrees of freedom. Empirical power was estimated as the proportion of replicates for which the null hypothesis was rejected. Source code is available at http://github.com/SampleSizeShop/mixedPower.

**4.1.1 Cluster randomized designs.** We compared approximate and empirical power for 36 cluster randomized designs. We assumed that each design had a single Gaussian outcome. Half of the clusters were assumed to have complete data, with the remaining clusters assumed to have some amount of missing data. We varied the number of treatment groups, $t \in \{2, 4\}$, the number of clusters randomized to each treatment, $N_{treatment} \in \{10, 40\}$, the total number of participants in a complete cluster, $p \in \{5, 50\}$ and the ratio of the incomplete cluster size to the complete cluster size $s \in \{0.6, 0.8, 1\}$. We only included designs which met the assumption that $N_d > (q + p_d + 1)$ for all $R_d$.

For each design, we repeated the simulations for several intraclass correlation values $\rho \in$ {0.04, 0.1, 0.2, 0.5}, with

$$\mathbf{\Sigma}_{max} = 2 \times \{\mathbf{1}_p \mathbf{1}'_p \rho + \mathbf{I}_p(1 - \rho)\}. \tag{44}$$

The $\boldsymbol{\beta}$ matrix had the form

$$\beta = b \times \begin{bmatrix} 1 & 0 \end{bmatrix}' \tag{45}$$

for designs with 2 treatments and

$$\beta = b \times \begin{bmatrix} 1 & 0 & 0 & 0 \end{bmatrix}' \tag{46}$$

for designs with 4 treatments. The scale factor $b$ was selected so that the approximate power was roughly 0.2, 0.5 or 0.8. In each scenario, we calculated power for the null hypothesis of no difference among treatment groups at $\alpha = 0.05$. We used the Wald test with denominator degrees of freedom as described by Kenward and Roger [1].

**4.1.2 Longitudinal designs.** We calculated approximate and empirical power for 36 longitudinal study designs. Each design had 5 repeated measures and 50 participants per treatment group. We varied the number of treatment groups, $t \in$ {2, 4}, the pattern of missing data, either monotone (missing the 4th and 5th observations), or non-monotone (missing the 2nd and 4th observations), and the number of participants in each treatment group with some amount of missing data, $N_{incomplete} \in$ {0, 10, 20}. For observations within a given participant, we assumed a first-order auto-regressive correlation structure [12, p. 99], with $\rho = 0.4$ and $\sigma^2 = 1$. The $\boldsymbol{\beta}$ matrix had the form

$$\boldsymbol{\beta} = b \times \begin{bmatrix} 1 & \mathbf{0}'_9 \end{bmatrix}' \tag{47}$$

for designs with 2 treatments and

$$\boldsymbol{\beta} = b \times \begin{bmatrix} 1 & \mathbf{0}'_{19} \end{bmatrix}' \tag{48}$$

for designs with 4 treatments. The scale factor and hypothesis testing were as described for the cluster randomized designs with one exception: we calculated power for the null hypothesis of no time by treatment interaction.

**4.1.3 Performance criteria.** For each design, we computed the deviation as approximate power minus empirical power. We produced box plots summarizing the deviations overall, within all cluster randomized trials, and within all longitudinal designs. For the cluster randomized trials, we produced box plots stratified by the number of treatment groups, the cluster size, and the ratio of the incomplete cluster size to the complete cluster size. For the longitudinal designs, we produced box plots summarizing the deviations stratified by the number of treatment groups, the pattern of missing observations, and the number of incomplete independent sampling units per treatment.

Positive deviations indicated that the approximate power values were larger than the empirical power values. Negative deviations indicated that the approximate power values were smaller than the empirical power values.

## 4.2 Results

Fig 1 summarizes the deviations between the approximate and the empirical power values. The three box plots show results for all designs, for cluster randomized trials, and for longitudinal studies. Overall, the median deviation between the approximate and the empirical power

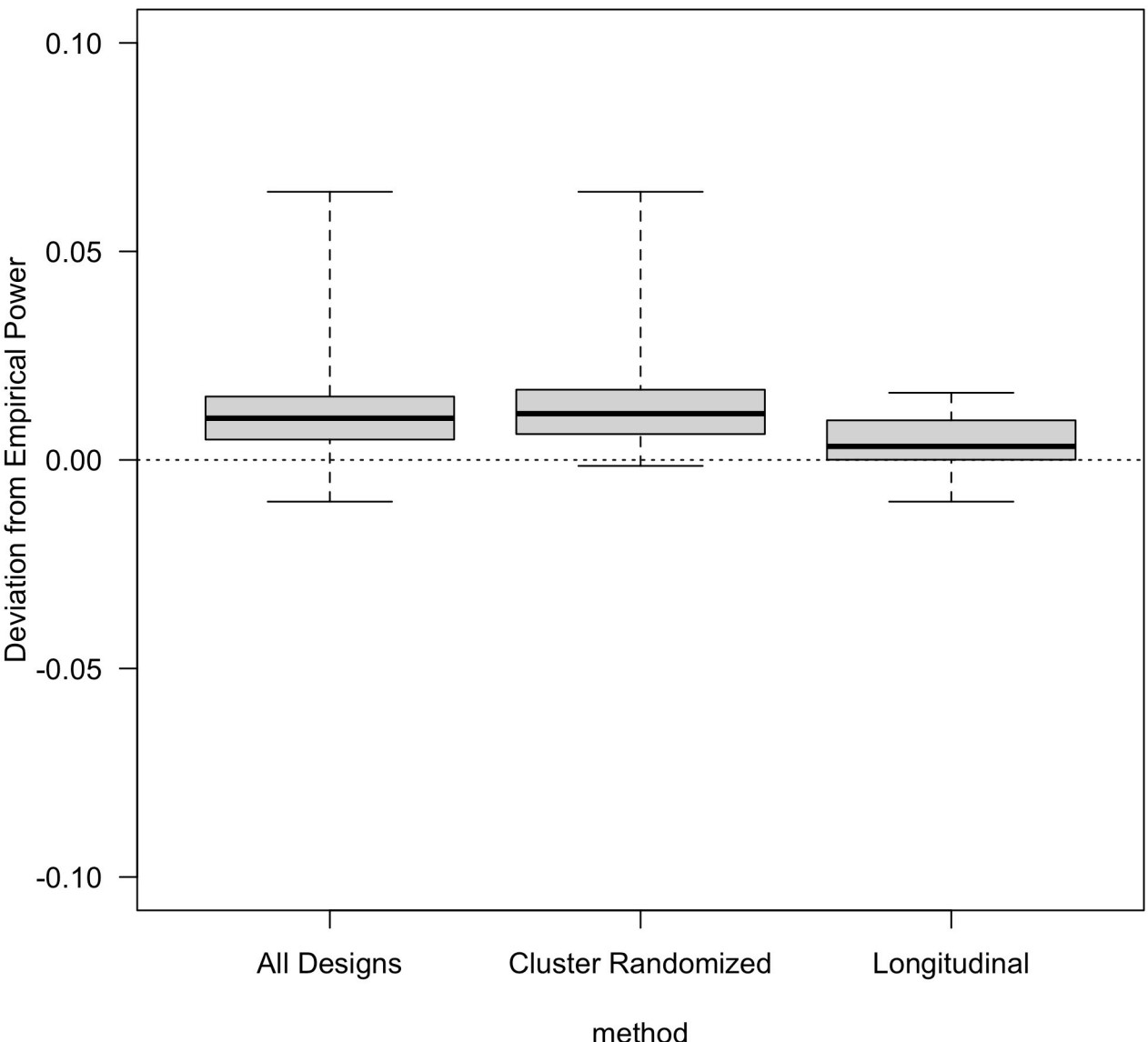

**Fig 1. Power deviations for all designs, cluster randomized designs only, and longitudinal designs only.** (center line, median; box limits, 1st and 3rd quartiles; whiskers, minimum and maximum).

values was 0.010 (min: −0.010, 1st quartile: 0.005, 3rd quartile: 0.015, max: 0.064). For cluster randomized trials, the median deviation was 0.011, (min: −0.001, 1st quartile: 0.006, 3rd quartile: 0.017, max: 0.064). For longitudinal designs, the median deviation was 0.003, (min: −0.010, 1st quartile: 0.000, 3rd quartile: 0.009, max: 0.016).

Further details for cluster-randomized designs are shown in Fig 2. The accuracy of the power approximation improved with larger cluster sizes. The approximation retained accuracy regardless of the ratio of incomplete to complete cluster sizes. As shown in Table 1, accuracy was similar across ICC values, with slight improvements with increasing correlation.

Results for longitudinal designs are shown in Fig 3. The power approximation was highly accurate for all longitudinal designs tested.

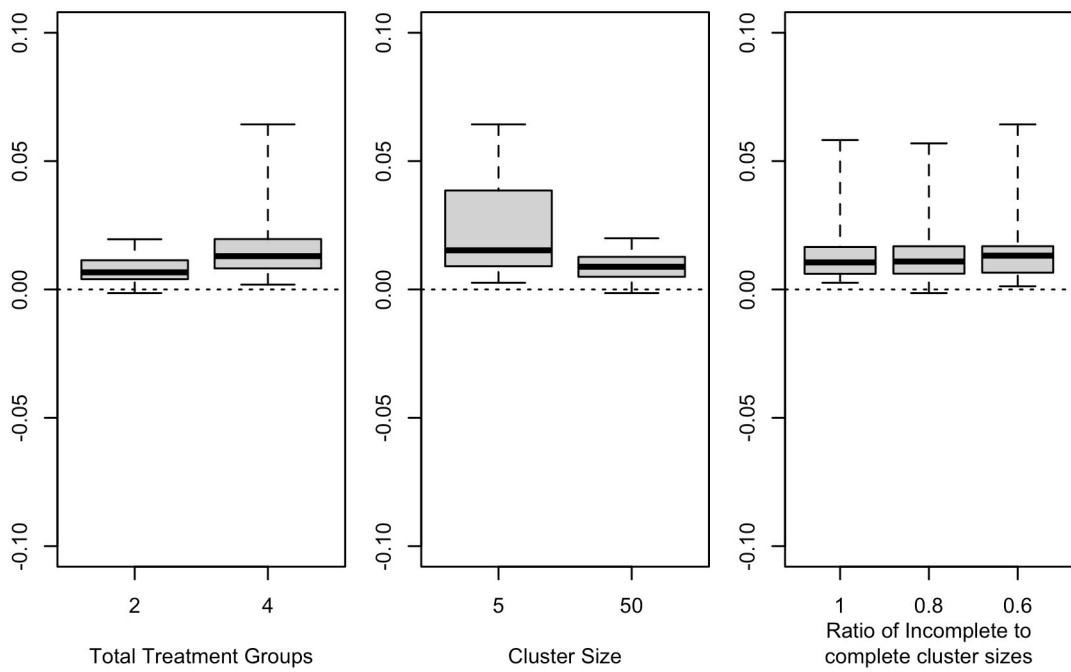

**Fig 2. Power deviations for cluster randomized designs.** (center line, median; box limits, 1st and 3rd quartiles; whiskers, minimum and maximum).

## 5 Applied example

We demonstrate a power calculation for an unbalanced cluster-randomized trial of an intervention to reduce oral cancer risk behaviors. The example is based on a hypothetical study examining the impact of workplace smoking cessation programs on tobacco use. We used a synthetic, rather than a real example, so that the power calculation is easy to follow. In a real power calculation, values of differences in means, standard deviations and intra-class correlation coefficients could be drawn from the literature, as described in Guo et al. [17].

For our demonstration, we assume that 80 worksites will be randomized to 2 smoking cessation programs, with 40 sites per treatment condition. Of the 40 sites randomized to each smoking cessation program, 25 worksites will have 30 participants, and the remaining 15 will have 20 participants. The outcome for the analysis will be urinary cotinine. We wish to detect a difference of 25 ng/ml. We assume a standard deviation of 125 ng/ml, and an intraclass correlation of 0.04. We will calculate power for the Kenward and Roger [1] test of the smoking cessation program effect. We set $\alpha = 0.05$.

To begin the calculation, we first identify the patterns of observations in the study, including complete clusters with 30 participants, and incomplete clusters with 20 participants.

**Table 1. Deviations between approximate and empirical power in cluster randomized designs by ICC.**

| ICC | Minimum | 1st Quartile | Median | 3rd Quartile | Maximum |
| --- | --- | --- | --- | --- | --- |
| 0.04 | -0.001 | 0.006 | 0.012 | 0.019 | 0.064 |
| 0.1 | 0.002 | 0.009 | 0.012 | 0.017 | 0.054 |
| 0.2 | 0.001 | 0.005 | 0.010 | 0.016 | 0.059 |
| 0.5 | 0.001 | 0.006 | 0.010 | 0.014 | 0.038 |

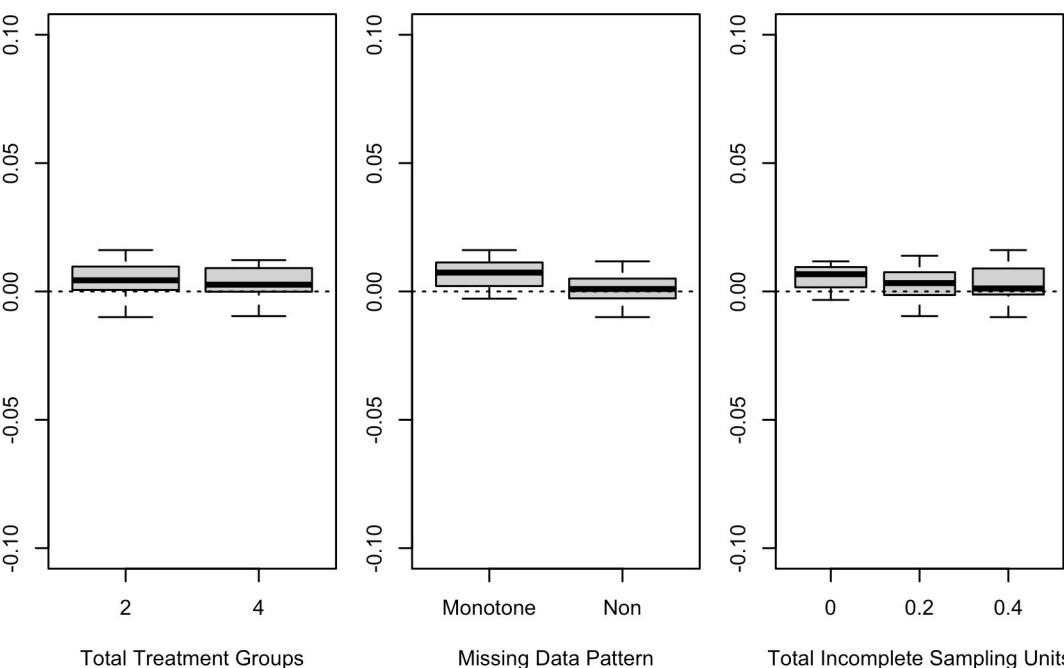

**Fig 3. Power deviations in longitudinal designs.** (center line, median; box limits, 1st and 3rd quartiles; whiskers, minimum and maximum).

Table 2 summarizes the design matrices and patterns of observations by cluster size and treatment assignment.

In addition, we define

$$\mathbf{\Sigma}_{max} = 125^2 \times \{\mathbf{1}_{30}\mathbf{1}'_{30} \times 0.04 + \boldsymbol{I}_{30}(1 - 0.04)\}, \tag{49}$$

$$\boldsymbol{C} = \begin{bmatrix} 1 & -1 \end{bmatrix}, \tag{50}$$

$$\boldsymbol{\theta}_0 = \begin{bmatrix} 0 \end{bmatrix} \tag{51}$$

and

$$\boldsymbol{\beta} = \begin{bmatrix} 25 & 0 \end{bmatrix}'. \tag{52}$$

At an $\alpha$ level of 0.05, the approximate power to detect a treatment difference of 25 ng/ml was 0.87 for the Wald test with Kenward and Roger [1] denominator degrees of freedom.

**Table 2. Design matrices and patterns of observations for proposed study of smoking cessation programs.**

|  | $p_i = 30$ | $p_i = 20$ |
|---|---|---|
| Program 1 | $\boldsymbol{X}_i = \mathbf{1}_{30} \otimes \begin{bmatrix} 1 & 0 \end{bmatrix}$ | $\boldsymbol{X}_i = \mathbf{1}_{20} \otimes \begin{bmatrix} 0 \end{bmatrix}$ |
|  | $R_d = \{1, \ldots, 30\}$ | $R_d = \{1, \ldots, 20\}$ |
| Program 2 | $\boldsymbol{X}_i = \mathbf{1}_{30} \otimes \begin{bmatrix} 1 & 1 \end{bmatrix}$ | $\boldsymbol{X}_i = \mathbf{1}_{20} \otimes \begin{bmatrix} 0 & 1 \end{bmatrix}$ |
|  | $R_d = \{1, \ldots, 30\}$ | $R_d = \{1, \ldots, 20\}$ |

## 6 Discussion

We describe a power approximation for the Kenward and Roger (1997) test of fixed effects in the linear mixed model. The method was accurate to within about ±0.06 for all designs, with the best accuracy observed for longitudinal designs. We note that Kenward and Roger (2009) have since described a refinement which improves estimation of the non-linear covariance structures in small samples. We have restricted our discussion to the Kenward and Roger (1997) approach, since it is most commonly used in statistical practice.

The method has several limitations. The assumption of $N_d > (q + p_d + 1)$ may be too restrictive for multilevel designs with large cluster sizes. In addition, we assume that the pattern of missing data is known. The method does not apply to repeated covariates, which often appear in biomedical studies. However, the method does apply to baseline covariates, a common study design. We make a strong homoscedasticity assumption of equal variance for each independent sampling unit. This assumption means that the power computations are not appropriate for random regression, for models with group differences in variance, or for certain spatial-temporal applications. Nevertheless, the assumption of homoscedasticity is widely made for randomized controlled clinical trials, laboratory studies, and observational studies, which makes the method useful for a variety of cases. Lastly, the method has not been evaluated for binary or Poisson data.

The analytic results from this manuscript suggest several future extensions. We may be able to calculate power for linear mixed models with random missing data patterns by invoking conditional distribution theory and calculating expected power across patterns of missingness. In addition, the approach used to form the distribution of $\hat{\boldsymbol{\Sigma}}_{s,M}$ provides the first step towards a non-iterative alternative to restricted maximum likelihood estimation for some mixed models. For big data applications, such a non-iterative approach may facilitate highly parallel computation of parameter estimates in mixed models.

Our power approximation provides a general, flexible, accurate and rapid method to calculate power for the Kenward and Roger (1997) test. For studies in which the Kenward and Roger (1997) test is the planned method of data analysis, our power approximation should be used. By aligning power analysis with the planned data analysis, researchers can more accurately assess power for biomedical studies. Accurate power analysis is an ethical imperative for research with human participants.

## 7 Appendix

## A Appendix: Theorems and proofs

**Theorem 1**. *For $m \in \{1, \ldots, k\}$, let $p_m \in \{1, 2, \ldots, p\}$, $N_m > (p_m + 3)$ and define $\boldsymbol{\Psi}_m = \{\psi_{mij}\}$ to be a $(p_m \times p_m)$ symmetric, positive definite matrix. Define a set of $k \geq 2$, independent, non-identically distributed inverse central Wishart random matrices, such that for $m \in \{1, \ldots, k\}$, $\boldsymbol{S}_m^{-1} \sim \mathcal{IW}_{p_m}(N_m, \boldsymbol{\Psi}_m)$. For $i \in \{1, \ldots, q\}$ and $R_m \subset \{1, 2, \ldots, p\}$ of cardinality $p_m$, define $\boldsymbol{X}_m$ to be a $(p_m \times qp)$ matrix of rank $p_m < qp$ with the form*

$$\boldsymbol{X}_m = \boldsymbol{I}_q(\{i\}) \otimes \boldsymbol{I}_p(R_m). \tag{53}$$

*If for each $i \in \{1, \ldots, q\}$, there exists at least one $m$ such that $\boldsymbol{X}_m = \boldsymbol{I}_q(\{i\}) \otimes \boldsymbol{I}_p$, then*

$$\boldsymbol{Q}^{-1} = \sum_{m=1}^{k} \boldsymbol{X}_m' \boldsymbol{S}_m^{-1} \boldsymbol{X}_m \tag{54}$$

*is positive definite.*

*Proof.* Let $Q_i = \{X_m: X_m = I_q(\{i\}) \otimes I_p(R_m)\}$. Then

$$\sum_{m=1}^{k} X_m' S_m^{-1} X_m$$

$$= \sum_{i=1}^{q} \sum_{Q_i} X_m' S_m^{-1} X_m$$

$$= \sum_{i=1}^{q} \sum_{Q_i} [I_q(\{i\}) \otimes I_p(R_m)]' S_m^{-1} [I_q(\{i\}) \otimes I_p(R_m)]$$

$$= \sum_{i=1}^{q} \sum_{Q_i} [I_q(\{i\})' \otimes I_p(R_m)'](I_1 \otimes S_m^{-1})[I_q(\{i\}) \otimes I_p(R_m)] \quad (55)$$

$$= \sum_{i=1}^{q} \sum_{Q_i} [I_q(\{i\})' \otimes I_p(R_m)' S_m^{-1}][I_q(\{i\}) \otimes I_p(R_m)]$$

$$= \sum_{i=1}^{q} \sum_{Q_i} [I_q(\{i\})' I_q(\{i\}) \otimes I_p(R_m)' S_m^{-1} I_p(R_m)]$$

$$= \sum_{i=1}^{q} \left\{ I_q(\{i\})' I_q(\{i\}) \otimes \sum_{Q_i} [I_p(R_m)' S_m^{-1} I_p(R_m)] \right\}.$$

Note that for $i \in \{1, 2, .., q\}$, $I_q(\{i\})' I_q(\{i\})$ is a $(q \times q)$ matrix for which the $i$th diagonal element is 1 and all remaining elements are 0. Therefore, Eq 55 can be equivalently expressed as a direct sum.

$$\sum_{m=1}^{k} X_m' S_m^{-1} X_m = \bigoplus_{i=1}^{q} \left\{ \sum_{Q_i} [I_p(R_m)' S_m^{-1} I_p(R_m)] \right\}. \quad (56)$$

From Mathai and Provost [18, p.18, Theorem 2.2b.1], it follows that each $I_p(R_m)' S_m^{-1} I_p(R_m)$ is positive semi-definite. By assumption, for each $Q_i$, there exists a $c_i$ such that $X_{c_i} \in Q_i$ such that $X_{c_i} = I_q(\{i\}) \otimes I_p$. Then

$$\sum_{m=1}^{k} X_m' S_m^{-1} X_m = \bigoplus_{i=1}^{q} \left\{ I_p S_{c_i}^{-1} I_p + \sum_{Q_i/X_{c_i}} [I_p(R_m)' S_m^{-1} I_p(R_m)] \right\}$$

$$= \bigoplus_{i=1}^{q} \left\{ S_{c_i}^{-1} + \sum_{Q_i/X_{c_i}} [I_p(R_m)' S_m^{-1} I_p(R_m)] \right\}. \quad (57)$$

Because $S_{c_i}^{-1}$ is positive definite and the remaining $I_p(R_m)' S_m^{-1} I_p(R_m)$ are positive semi-definite for $i \in \{1, \ldots, q\}$, then

$$S_{c_i}^{-1} + \sum_{Q_i/X_{c_i}} [I_p(R_m)' S_m^{-1} I_p(R_m)] \quad (58)$$

is positive definite.

Since $\sum_{m=1}^{k} X_m' S_m^{-1} X_m$ is a block matrix, the eigenvalues of $\sum_{m=1}^{k} X_m' S_m^{-1} X_m$ are the eigenvalues of all of the blocks. Since each block (Eq 58) is positive definite and hence has positive eigenvalues, it follows that $\sum_{m=1}^{k} X_m' S_m^{-1} X_m$ must also be positive definite.

**Theorem 2**. *For $m \in \{1, \ldots, k\}$, $i \in \{1, \ldots, q\}$, $R_m \subset \{1, 2, \ldots, p\}$ of cardinality $p_m$, $X_m = I_q(\{i\}) \otimes I_p(R_m)$ a $(p_m \times qp)$ matrix of rank $p_m < qp$, $N_m > (p_m + 3)$, $\Psi_m$ a $(p_m \times p_m)$ symmetric, positive definite matrix, and $S_m^{-1} \sim \mathcal{IW}_{p_m}(N_m, \Psi_m)$,*

$$\text{tr}(S_m^{-1}) = \text{tr}(X_m' S_m^{-1} X_m). \tag{59}$$

*Proof.* Let $\text{Dg}(x)$ indicate a square matrix with the elements of the vector $x$ on the diagonal.

Since $S_m^{-1}$ is positive definite and has full rank, then by Lemma 1.24 (a) of Muller and Stewart [9], it has the spectral decomposition

$$S_m^{-1} = V\text{Dg}(\lambda)V', \tag{60}$$

where $\lambda$ is the $(p_m \times 1)$ vector of eigenvalues and $V$ is the $(p_m \times p_m)$ orthogonal matrix of eigenvectors of $S_m^{-1}$. Then

$$X_m' S_m^{-1} X_m = X_m' V\text{Dg}(\lambda)V' X_m. \tag{61}$$

Since $X_m$ has deficient rank $p_m < qp$, then by Lemma 1.25 of of Muller and Stewart [9] it must have $qp - p_m$ zero eigenvalues. Let $\lambda_0$ be the $(qp - p_m \times 1)$ vector of zero eigenvalues and $V_0$ the $[qp \times (qp - p_m)]$ matrix of corresponding eigenvectors. Then

$$
\begin{aligned}
X_m' S_m^{-1} X_m &= X_m' V\text{Dg}(\lambda)V' X_m \\
&= X_m' V\text{Dg}(\lambda)V' X_m + V_0\text{Dg}(\lambda_0)V_0' \\
&= \begin{bmatrix} X_m' V & V_0 \end{bmatrix} \begin{bmatrix} \text{Dg}(\lambda) & 0 \\ 0 & \text{Dg}(\lambda_0) \end{bmatrix} \begin{bmatrix} V' X_m \\ V_0' \end{bmatrix}.
\end{aligned}
\tag{62}
$$

Selecting $V_0$ such that $V_0' V_0 = I_{qp-p_m}$, $V_0 V_0' = I - X_m' X_m$ and $X_m V_0 = 0$, ensures that $\begin{bmatrix} X_m' V & V_0 \end{bmatrix}$ is orthogonal. Then Eq 62 is the spectral decomposition of $X_m' S_m^{-1} X_m$, with eigenvalues $\begin{bmatrix} \lambda' & \lambda_0' \end{bmatrix}'$.

Since $\lambda_0$ contains only zero eigenvalues and using the definition of the trace,

$$
\begin{aligned}
\text{tr}(X_m' S_m^{-1} X_m) &= \sum_{i=1}^{p_m} \lambda_i + \sum_{j=1}^{qp-p_m} \lambda_{0j} \\
&= \sum_{i=1}^{p_m} \lambda_i \\
&= \text{tr}(S_m^{-1}).
\end{aligned}
\tag{63}
$$

**Theorem 3**. *For $m \in \{1, \ldots, k\}$, let $p_m \in \{1, \ldots, p\}$, $N_m > (p_m + 3)$ and let $\Psi_m = \{\psi_{mij}\}$ be a $(p_m \times p_m)$ symmetric, positive definite matrix. Define a set of $k \geq 2$, independent, non-identically distributed inverse central Wishart random matrices, such that for $m \in \{1, \ldots, k\}$, $S_m^{-1} \sim \mathcal{IW}_{p_m}(N_m, \Psi_m)$. For $i \in \{1, \ldots, q\}$ and $R_m \subset \{1, \ldots, p\}$ of cardinality $p_m$, define $X_m$ to be a $(p_m \times qp)$ matrix of rank $p_m < qp$ with the form*

$$X_m = I_q(\{i\}) \otimes I_p(R_m). \tag{64}$$

*Under the assumption that for each $i \in \{1, \ldots, q\}$, there exists at least one $m$ such that*

$X_m = I_q(\{i\}) \otimes I_p$, it can be shown that

$$Q^{-1} = \sum_{m=1}^{k} X'_m S_m^{-1} X_m \tag{65}$$

is approximately distributed as $S_*^{-1} \sim \mathcal{IW}_{qp}(N_*, \Psi_*)$.

*Proof.* Theorem 1 in Appendix demonstrates that $Q^{-1}$ is positive definite under the restriction that for each $i \in \{1, \ldots, q\}$, there exists at least one $m$ such that $X_m = I_q(\{i\}) \otimes I_p$.

To derive an approximate distribution for $Q^{-1}$, we match the expectation of the sum of the Wishart matrices and the variance of the trace of the sum of the Wishart matrices. Set

$$E(S_*^{-1}) = E\left( \sum_{m=1}^{k} X'_m S_m^{-1} X_m \right) \tag{66}$$

and

$$\mathcal{V}[\mathrm{tr}(S_*^{-1})] = \mathcal{V}\left[ \mathrm{tr}\left( \sum_{m=1}^{k} X'_m S_m^{-1} X_m \right) \right]. \tag{67}$$

From Theorem 2 in Appendix and the independence of the $S_m^{-1}$,

$$\mathcal{V}\left[ \mathrm{tr}\left( \sum_{m=1}^{k} X'_m S_m^{-1} X_m \right) \right] = \sum_{m=1}^{k} \mathcal{V}[\mathrm{tr}(X'_m S_m^{-1} X_m)]$$

$$= \sum_{m=1}^{k} \mathcal{V}[\mathrm{tr}(S_m^{-1})]. \tag{68}$$

Then the approximate parameters for $S_*^{-1} \sim \mathcal{IW}_q(N_*, \Psi_*)$ are

$$\Psi_* = (N_* - q - 1) \sum_{m=1}^{k} \frac{1}{N_m - p_m - 1} X'_m \Psi_m X_m \tag{69}$$

and

$$N_* = \frac{b_h + \sqrt{b_h^2 - 4h_4 c_h}}{2h_4}, \tag{70}$$

where

$$h_1 = \sum_{i=1}^{qp} \left[ \sum_{m=1}^{k} \frac{1}{N_m - p_m - 1} (X'_m \Psi_m X_m)_{ii} \right]^2, \tag{71}$$

$$h_2 = \sum_{1 < i < j < qp} \left\{ \left[ \sum_{m=1}^{k} \frac{1}{N_m - p_m - 1} (X'_m \Psi_m X_m)_{ii} \right] \times \left[ \sum_{m=1}^{k} \frac{1}{N_m - p_m - 1} \left( X'_m \Psi_m X_m \right)_{jj} \right] \right\}, \tag{72}$$

$$h_3 = \sum_{1 < i < j < qp} \left[ \sum_{m=1}^{k} \frac{1}{N_m - p_m - 1} (X'_m \Psi_m X_m)_{ij} \right]^2, \tag{73}$$

$$h_4 \;=\; \sum_{m=1}^{k}\sum_{i=1}^{p_m}\frac{2\psi_{mii}^2}{(N_m-p_m-1)^2(N_m-p_m-3)}$$
$$+4\sum_{m=1}^{k}\sum_{1<i<j<p_m}\frac{\psi_{mii}\psi_{mjj}+(N_m-p_m-1)\psi_{mij}^2}{(N_m-p_m)(N_m-p_m-1)^2(N_m-p_m-3)}, \tag{74}$$

$$b_h = (2h_1 + 4h_3 + 2h_4 p + 3h_4), \tag{75}$$

and

$$c_h \;=\; (2h_1 p - 4h_2 + 4h_3 p + 4h_3 + h_4 p^2 + 3h_4 p). \tag{76}$$

The method of moments approximation yields an asymptotic approximation for the sum, as desired.

**Theorem 4.** *Let n and p be positive integers, $\boldsymbol{\mu}$ be a ($p \times 1$) vector of means, and $\boldsymbol{\Sigma}_x \neq \boldsymbol{\Sigma}_W$ be symmetric and positive definite ($p \times p$) matrices. Suppose $\boldsymbol{x} \sim \mathcal{N}_p(\boldsymbol{\mu}, \boldsymbol{\Sigma}_x)$ independently of $\boldsymbol{W} \sim \mathcal{W}_p(n, \boldsymbol{\Sigma}_W)$. Then*

$$\boldsymbol{x}'\boldsymbol{W}^{-1}\boldsymbol{x} \;\dot\sim\; \frac{\lambda_u n_u}{(n+p-1)}\mathcal{F}\{n_u, (n+p-1), \delta_u\}, \tag{77}$$

*with*

$$\lambda_u = \delta_u^{-1}(\boldsymbol{\mu}'\boldsymbol{\Sigma}_W^{-1}\boldsymbol{\mu}), \tag{78}$$

$$n_u = \delta_u(\boldsymbol{\mu}'\boldsymbol{\Sigma}_W^{-1}\boldsymbol{\mu})^{-1}\mathrm{tr}(\boldsymbol{\Sigma}_W^{-1}\boldsymbol{\Sigma}_x), \tag{79}$$

*and*

$$\delta_u = \frac{(\boldsymbol{\mu}'\boldsymbol{\Sigma}_W^{-1}\boldsymbol{\mu})\mathrm{tr}(\boldsymbol{\Sigma}_W^{-1}\boldsymbol{\Sigma}_x) + 2(\boldsymbol{\mu}'\boldsymbol{\Sigma}_W^{-1}\boldsymbol{\mu})^2}{\mathrm{tr}(\boldsymbol{\Sigma}_W^{-1}\boldsymbol{\Sigma}_x\boldsymbol{\Sigma}_W^{-1}\boldsymbol{\Sigma}_x) + 2\boldsymbol{\mu}'\boldsymbol{\Sigma}_W^{-1}\boldsymbol{\Sigma}_x\boldsymbol{\Sigma}_W^{-1}\boldsymbol{\mu}}. \tag{80}$$

*Proof.* Define $V = \boldsymbol{x}'\boldsymbol{\Sigma}_W^{-1}\boldsymbol{x}/\boldsymbol{x}'\boldsymbol{W}^{-1}\boldsymbol{x}$. Define $U = \boldsymbol{x}'\boldsymbol{\Sigma}_W^{-1}\boldsymbol{x}$. Using Lemma 17.10 in Arnold [19, p. 319], it follows that $V|\boldsymbol{x} \sim \chi^2_{n+p-1}$. Hence, $V \perp \boldsymbol{x}$, which implies $V \perp U$.

The expression $U$ is a weighted sum of noncentral $\chi^2$ random variables [9, Theorem 9.5, p. 176]. Approximate the distribution of $U$ with a single noncentral $\chi^2$, so that $U \dot\sim \lambda_u \chi^2_{n_u}(\delta_u)$. Using the approach described by Kim et al. [8], obtain values for $\lambda_u$, $n_u$ and $\delta_u$ by matching the following three moments:

$$E_0\{\lambda_u\chi^2_{n_u}(\delta_u)\} = E_0(U), \tag{81}$$

$$E_A\{\lambda_u\chi^2_{n_u}(\delta_u)\} = E_A(U), \tag{82}$$

and

$$\mathcal{V}_A\{\lambda_u\chi^2_{n_u}(\delta_u)\} = \mathcal{V}_A(U). \tag{83}$$

The moments of $U$ are [9, Corollary 9.6.3, p. 179],

$$E_0(U) = \text{tr}(\boldsymbol{\Sigma}_W^{-1}\boldsymbol{\Sigma}_x), \tag{84}$$

$$E_A(U) = \text{tr}(\boldsymbol{\Sigma}_W^{-1}\boldsymbol{\Sigma}_x) + \boldsymbol{\mu}'\boldsymbol{\Sigma}_W^{-1}\boldsymbol{\mu}, \tag{85}$$

and

$$\mathcal{V}_A(U) = 2\text{tr}(\boldsymbol{\Sigma}_W^{-1}\boldsymbol{\Sigma}_x\boldsymbol{\Sigma}_W^{-1}\boldsymbol{\Sigma}_x) + 4\boldsymbol{\mu}'\boldsymbol{\Sigma}_W^{-1}\boldsymbol{\Sigma}_x\boldsymbol{\Sigma}_W^{-1}\boldsymbol{\mu}. \tag{86}$$

Then the approximate parameters of $U$ are

$$\lambda_u = \delta_u^{-1}(\boldsymbol{\mu}'\boldsymbol{\Sigma}_W^{-1}\boldsymbol{\mu}), \tag{87}$$

$$n_u = \delta_u(\boldsymbol{\mu}'\boldsymbol{\Sigma}_W^{-1}\boldsymbol{\mu})^{-1}\text{tr}(\boldsymbol{\Sigma}_W^{-1}\boldsymbol{\Sigma}_x), \tag{88}$$

and

$$\delta_u = \frac{(\boldsymbol{\mu}'\boldsymbol{\Sigma}_W^{-1}\boldsymbol{\mu})\text{tr}(\boldsymbol{\Sigma}_W^{-1}\boldsymbol{\Sigma}_x) + 2(\boldsymbol{\mu}'\boldsymbol{\Sigma}_W^{-1}\boldsymbol{\mu})^2}{\text{tr}(\boldsymbol{\Sigma}_W^{-1}\boldsymbol{\Sigma}_x\boldsymbol{\Sigma}_W^{-1}\boldsymbol{\Sigma}_x) + 2\boldsymbol{\mu}'\boldsymbol{\Sigma}_W^{-1}\boldsymbol{\Sigma}_x\boldsymbol{\Sigma}_W^{-1}\boldsymbol{\mu}}. \tag{89}$$

Since $(U/\lambda_u) \;\dot\sim\; \chi_{n_u}^2(\delta_u)$, $V \sim \chi_{n+p-1}^2$, and $V \perp U$,

$$\frac{U/(\lambda_u n_u)}{V/(n+p-1)} \;\dot\sim\; \mathcal{F}\{n_u, (n+p-1), \delta_u\}.$$

Because $U/V = \boldsymbol{x}'\boldsymbol{W}^{-1}\boldsymbol{x}$,

$$(\boldsymbol{x}'\boldsymbol{W}^{-1}\boldsymbol{x})\frac{(n+p-1)}{\lambda_u n_u} \;\dot\sim\; \mathcal{F}\{n_u, (n+p-1), \delta_u\},$$

and the result follows.

## Acknowledgments

A portion of this paper was submitted to the University of Colorado Denver in partial fulfillment of the requirements for the degree of Doctor of Philosophy in Biostatistics for Dr. Sarah M. Kreidler.

## Author Contributions

**Conceptualization:** Sarah M. Kreidler, Keith E. Muller, Deborah H. Glueck.

**Data curation:** Sarah M. Kreidler.

**Formal analysis:** Sarah M. Kreidler.

**Funding acquisition:** Keith E. Muller, Deborah H. Glueck.

**Investigation:** Sarah M. Kreidler, Deborah H. Glueck.

**Methodology:** Sarah M. Kreidler, Keith E. Muller, Deborah H. Glueck.

**Project administration:** Sarah M. Kreidler, Keith E. Muller, Deborah H. Glueck.

**Resources:** Keith E. Muller, Deborah H. Glueck.

**Software:** Sarah M. Kreidler.

**Supervision:** Sarah M. Kreidler, Keith E. Muller, Deborah H. Glueck.

**Validation:** Sarah M. Kreidler, Deborah H. Glueck.

**Visualization:** Sarah M. Kreidler.

**Writing – original draft:** Sarah M. Kreidler, Deborah H. Glueck.

**Writing – review & editing:** Sarah M. Kreidler, Brandy M. Ringham, Keith E. Muller, Deborah H. Glueck.

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
