## [Decision Letter · Decision Letter 0]

3 Mar 2021

PONE-D-21-00404

A power approximation for the Kenward and Roger Wald test in the linear mixed model

PLOS ONE

Dear Dr. Ringham

Thank you for submitting your manuscript to PLOS ONE. After careful consideration, we feel that it has merit but does not fully meet PLOS ONE’s publication criteria as it currently stands. Therefore, we invite you to submit a revised version of the manuscript that addresses the points raised during the review process.

We look forward to receiving your revised manuscript.

Kind regards,

Lei Shi

Academic Editor

PLOS ONE

Journal Requirements:

We note that one or more of the authors are employed by a commercial company: Sunrun Inc.

2.1. Please provide an amended Funding Statement declaring this commercial affiliation, as well as a statement regarding the Role of Funders in your study. If the funding organization did not play a role in the study design, data collection and analysis, decision to publish, or preparation of the manuscript and only provided financial support in the form of authors' salaries and/or research materials, please review your statements relating to the author contributions, and ensure you have specifically and accurately indicated the role(s) that these authors had in your study. You can update author roles in the Author Contributions section of the online submission form.

2.2. Please also provide an updated Competing Interests Statement declaring this commercial affiliation along with any other relevant declarations relating to employment, consultancy, patents, products in development, or marketed products, etc.  

Funding for this work was provided by NIDCR 1 R01 DE020832-01A1 (Keith E. Muller,

PI; Deborah H. Glueck, University of Colorado site PI), NIGMS 9R01GM121081-05

(Deborah H. Glueck, Keith E. Muller, Dana Dabelea, PIs), and OD 5UG3OD023248-02 

(Dana Dabelea, PI).

"KEM, DHG

NIDCR 1 R01 DE020832-01A1

The National Institute of Dental and Craniofacial Research

www.nih.gov

No

KEM, DHG

NIGMS 9R01GM121081-05

The National Institute of General Medical Sciences

www.nih.gov

No

Reviewers' comments:

Reviewer's Responses to Questions

**Comments to the Author**

1. Is the manuscript technically sound, and do the data support the conclusions?

Reviewer #1: Yes

Reviewer #2: Yes

2. Has the statistical analysis been performed appropriately and rigorously? 

Reviewer #1: Yes

Reviewer #2: Yes

3. Have the authors made all data underlying the findings in their manuscript fully available?

Reviewer #1: Yes

Reviewer #2: Yes

4. Is the manuscript presented in an intelligible fashion and written in standard English?

Reviewer #1: Yes

Reviewer #2: Yes

5. Review Comments to the Author

Reviewer #1: In this paper, a power approximation of Kenward and Roger test is derived. Via Monte Carlo simulation, author’s demonstrate that the new power approximation is accurate for cluster randomised trials and longitudinal study designs. The paper is well written and addresses an interesting problem.

My major issues are listed below.Major comments:

Comment 1: On line 171, it is claimed that if Σs and β are estimated using multivariate techniques, independence would follow. Provide a reference for this or give a detailed explanation in support of this claim.

Comment 2: The comparison of empirical and proposed powers is done assuming intraclass correlation(ICC) 0.04. This value of ICC is very small and in practice it can vary up to 0.5. Make the comparison of powers for higher values of ICC as well (say 0.1, 0.2, 0.5).

Comment 3: In the spirit of the longitudinal studies, how efficient is the power approximation when the correlation structure is assumed to be auto–regressive?

Comment 4: In Section 5 (Applied Example), rather than assuming the values of standard deviation and intraclass correlation, it is more reasonable to use the estimates of the parameters obtained from the data.

Minor comments:

Comment 1:Check line 122

Reviewer #2: The authors here present a noncentral F power approximation for the Kenward and Roger test. This work is innovative, and the organization of the manuscript is clear and comprehensive. Below are some minor comments that could help further streamline the text.

My review comments has been uploaded as an attachment.

6. PLOS authors have the option to publish the peer review history of their article (what does this mean?). If published, this will include your full peer review and any attached files.

Reviewer #1: No

Reviewer #2: **Yes: **Huang Lin

---

## [Author Response · Author response to Decision Letter 0]

10 May 2021

Response to Reviewers

We thank the reviewers for their kind comments. We have responded to all comments below. Reviewer comments are in italics and our response is in plain type.

General Comments

Done.

"The authors have declared that no competing interests exist." We note that one or more of the authors are employed by a commercial company: Sunrun Inc. Please provide an amended Funding Statement declaring this commercial affiliation, as well as a statement regarding the Role of Funders in your study. If the funding organization did not play a role in the study design, data collection and analysis, decision to publish, or preparation of the manuscript and only provided financial support in the form of authors' salaries and/or research materials, please review your statements relating to the author contributions, and ensure you have specifically and accurately indicated the role(s) that these authors had in your study. You can update author roles in the Author Contributions section of the online submission form.

Please also provide an updated Competing Interests Statement declaring this commercial affiliation along with any other relevant declarations relating to employment, consultancy, patents, products in development, or marketed products, etc.

We have removed Dr. Kreidler’s affiliation with SunRun, Inc. Previously we indicated that Dr. Kreidler was affiliated with SunRun Inc. After further review, and examining the policies of PLOS One, we realized that SunRun did not play any role in the study design, data collection and analysis, decision to publish, or preparation of the manuscript and, in addition, did not provide support in the form of salaries for any author, including SMK. 

We have updated Dr. Kreidler’s affiliation to be the Department of Biostatistics and Information, University of Colorado Denver. Although Dr. Kreidler is currently employed by SunRun, Inc., all of the work on this manuscript was completed while Dr. Kreidler was a doctoral student at the University of Colorado Denver. Current revisions are being done during Dr. Kreidler’s personal time. No SunRun, Inc. resources were used in the revisions of this manuscript for submission. The publication of this manuscript will not affect SunRun Inc., nor Dr. Kreidler in any financial way.

3. Thank you for stating the following in the Acknowledgments Section of your manuscript: Funding for this work was provided by NIDCR 1 R01 DE020832-01A1 (Keith E. Muller, PI; Deborah H. Glueck, University of Colorado site PI), NIGMS 9R01GM121081-05 (Deborah H. Glueck, Keith E. Muller, Dana Dabelea, PIs), and OD 5UG3OD023248-02 (Dana Dabelea, PI).

"KEM, DHG

NIDCR 1 R01 DE020832-01A1

The National Institute of Dental and Craniofacial Research

www.nih.gov

No

KEM, DHG

NIGMS 9R01GM121081-05

The National Institute of General Medical Sciences

www.nih.gov

No

We have provided an amended funding statement in the cover letter for this resubmission.

 

Reviewer 1 Comments

1. On line 171, it is claimed that if Σs and β are estimated using multivariate techniques, independence would follow. Provide a reference for this or give a detailed explanation in support of this claim.

We now provide a citation to support the statement above where it is mentioned in the manuscript. For reference, the citation is Anderson (1984, pg. 291, Theorem 8.2.2).

2. The comparison of empirical and proposed powers is done assuming intraclass correlation(ICC) 0.04. This value of ICC is very small and in practice it can vary up to 0.5. Make the comparison of powers for higher values of ICC as well (say 0.1, 0.2, 0.5).

We have re-run the simulation after incorporating the reviewer’s suggestions. The new results appear in the revised manuscript. The different ICC’s do not appear to change the accuracy of the results.

3. In the spirit of the longitudinal studies, how efficient is the power approximation when the correlation structure is assumed to be auto–regressive?

We used an auto-regressive correlation structure for the simulation studies described in Section 4.1.2 Longitudinal Designs. The median deviation between the approximate power and the true power was 0.003 (range: -0.010 to 0.016; 1st quartile: 0.00, 3rd quartile: 0.009).

4. In Section 5 (Applied Example), rather than assuming the values of standard deviation and intraclass correlation, it is more reasonable to use the estimates of the parameters obtained from the data.

The example is a synthetic example designed to demonstrate the utility of the calculations for a simple, cluster randomized study with different sized clusters. While the example was inspired by the trial described by Hennrikus et al., the example was so simplified that the reference to the Hennrikus et al. paper was confusing, rather than helpful. We have removed the reference to Hennrikus et al., clarified that the example is synthetic, and added a sentence to describe how a researcher might extract the inputs for the power analysis from review of the literature.

5. Check line 122

We reviewed the line and removed the duplicated equation. Thank you for finding the error. 

Reviewer 2 Comments

1. The homoscedasticity assumption (line 136) the author made could be a key limitation of the approximation method with the presence of different random coefficients. Even though it has been mentioned that there are no repeated covariates (line 82), it is worth mentioning it in the Discussion section.

We now describe limitations due to model assumptions in the Discussion section.

“The method does not apply to repeated covariates, which often appear in biomedical studies. However, the method does apply to baseline covariates, a common study design. We make a strong homoscedasticity assumption of equal variance for each independent sampling unit. This assumption means that the power computations are not appropriate for random regression, for models with group differences in variance, or for certain spatial-temporal applications. Nevertheless, the assumption of homoscedasticity is widely made for randomized controlled clinical trials, laboratory studies, and observational studies, which makes the method useful for a variety of cases.”

2. Theorem 3 in the Appendix lacks the reference regarding “the sum of the inverse Wishart distribution asymptotically or approximately follows an inverse Wishart distribution.”

We have now clarified the proof for Theorem 3 to explicitly state what we set out to prove. We added the following text to the end of the proof.

“The method of moments approximation yields an asymptotic approximation for the sum, as desired.”

3. Using p_i to denote the number of observations is counter-intuitive, especially given you define n=∑_(i=1)^N▒p_i . Replacing p_i with n_i would be easier to read.

We recognize the value of using n to denote different sample sizes. However, for this particular application, we use multivariate modeling theory to develop our results. In multivariate notation, it is standard to refer to the repeated measures using p. When the data are stacked, the vector of observations for independent sampling unit i is then given pi. We recognize that the notation may appear slightly awkward, but respectfully request that we may keep it as written so that it aligns with standard notation in the field. See, for example, Muller and Stuart (2006).

4. Duplicated terms appeared in line 122.

Thank you for finding the error. We have corrected it.

5. The notation for noncentrality parameter of F distribution is very similar to the notation of Wald statistic, please consider changing one of them.

We agree and have now adjusted the notation for the non-centrality parameter that appears in Section 2.1 Notation.

6. Box-plot elements should be defined (e.g. center line, median; box limits, upper and lower quartiles; whiskers, 1.5x interquartile range; points, outliers) in the legend.

Done.

---

## [Decision Letter · Decision Letter 1]

5 Jul 2021

A power approximation for the Kenward and Roger Wald test in the linear mixed model

PONE-D-21-00404R1

Dear Dr. Brandy Ringham,

We’re pleased to inform you that your manuscript has been judged scientifically suitable for publication and will be formally accepted for publication once it meets all outstanding technical requirements.

Kind regards,

Lei Shi

Academic Editor

PLOS ONE

Additional Editor Comments (optional):

Reviewers' comments:

Reviewer's Responses to Questions

**Comments to the Author**

1. If the authors have adequately addressed your comments raised in a previous round of review and you feel that this manuscript is now acceptable for publication, you may indicate that here to bypass the “Comments to the Author” section, enter your conflict of interest statement in the “Confidential to Editor” section, and submit your "Accept" recommendation.

Reviewer #1: All comments have been addressed

Reviewer #2: All comments have been addressed

2. Is the manuscript technically sound, and do the data support the conclusions?

Reviewer #1: Yes

Reviewer #2: (No Response)

3. Has the statistical analysis been performed appropriately and rigorously? 

Reviewer #1: Yes

Reviewer #2: (No Response)

4. Have the authors made all data underlying the findings in their manuscript fully available?

Reviewer #1: Yes

Reviewer #2: (No Response)

5. Is the manuscript presented in an intelligible fashion and written in standard English?

Reviewer #1: Yes

Reviewer #2: (No Response)

6. Review Comments to the Author

Reviewer #1: In the revised version authors successfully incorporated all the suggestions and corrections.

All comments are addressed adequately.

Reviewer #2: (No Response)

7. PLOS authors have the option to publish the peer review history of their article (what does this mean?). If published, this will include your full peer review and any attached files.

Reviewer #1: No

Reviewer #2: **Yes: **Huang Lin

---

## [Editor Report · Acceptance letter]

9 Jul 2021

PONE-D-21-00404R1 

A Power Approximation for the Kenward and Roger Wald Test in the Linear Mixed Model 

Dear Dr. Ringham:

I'm pleased to inform you that your manuscript has been deemed suitable for publication in PLOS ONE. Congratulations! Your manuscript is now with our production department. 

Kind regards, 

on behalf of

Dr. Lei Shi 

Academic Editor

PLOS ONE